# Modification of Critical Current Density Anisotropy in High-$T_c$ Superconductors by Using Heavy-Ion Irradiations

Tetsuro Sueyoshi

Department of Electrical Engineering, Kyushu Sangyo University, 2-3-1 Matsukadai Higashi-ku, Fukuoka 813-8503, Japan; s.teturo@ip.kyusan-u.ac.jp; Tel.: +81-92-673-5632; Fax: +81-92-673-5091

**Abstract:** The critical current density $J_c$, which is a maximum value of zero-resistivity current density, is required to exhibit not only larger value but also lower anisotropy in a magnetic field $B$ for applications of high-$T_c$ superconductors. Heavy-ion irradiation introduces nanometer-scale irradiation tracks, i.e., columnar defects (CDs) into high-$T_c$ superconducting materials, which can modify both the absolute value and the anisotropy of $J_c$ in a controlled manner: the unique structures of CDs, which significantly affect the $J_c$ properties, are engineered by adjusting the irradiation conditions such as the irradiation energy and the incident direction. This paper reviews the modifications of the $J_c$ anisotropy in high-$T_c$ superconductors using CDs installed by heavy-ion irradiations. The direction-dispersion of CDs, which is tuned by the combination of the plural irradiation directions, can provide a variety of the magnetic field angular variations of $J_c$ in high-$T_c$ superconductors: CDs crossing at $\pm\theta_i$ relative to the $c$-axis of YBa$_2$Cu$_3$O$_y$ films induce a broad peak of $J_c$ centered at $B \parallel c$ for $\theta_i < \pm 45°$, whereas the crossing angle of $\theta_i \geq \pm 45°$ cause not a $J_c$ peak centered at $B \parallel c$ but two peaks of $J_c$ at the irradiation angles. The anisotropy of $J_c$ can also modified by tuning the continuity of CDs: short segmented CDs formed by heavy-ion irradiation with relatively low energy are more effective to improve $J_c$ in a wide magnetic field angular region. The modifications of the $J_c$ anisotropy are discussed on the basis of both structures of CDs and flux line structures depending on the magnetic field directions.

**Keywords:** high-$T_c$ superconductors; critical current density; flux pinning; heavy-ion irradiation; columnar defects; anisotropy

## 1. Introduction

High-$T_c$ superconductors have attracted considerable research activity, especially for electric power applications at high magnetic fields and temperatures, because the zero-resistive current and the high superconducting transition temperature $T_c$ enable us to operate zero-resistance devices at liquid-nitrogen temperature. Nowadays, coated conductors based on biaxially textured REBa$_2$Cu$_3$O$_y$ (REBCO, RE: rare earth elements) thin films have been significantly developed as second generation high-$T_c$ superconducting tapes and have become commercially available now [1,2].

The critical current density $J_c$ in magnetic field (in-field $J_c$), which is a maximum current density with zero-resistivity, is the most important parameter in REBCO-coated conductors for the practical applications. The absolute values of $J_c$ for REBCO-coated conductors, however, have still remained below the practical level for high magnetic field applications [3]. In addition, the electronic mass anisotropy in the layered structure of CuO$_2$ planes for high-$T_c$ superconductors induces a large anisotropy of $J_c$ against a magnetic field orientation [4], which gives rise to obstacles to the superconducting magnet applications: a minimum in the magnetic field angular variation of $J_c$, which is usually located at the magnetic field $B$ parallel to the $c$-axis, limits the operation current [5,6].

The in-field $J_c$ can be controlled by immobilization of nano-sized quantized-magnetic-flux-lines (flux lines) penetrating into superconductors in a magnetic field. The motion of

flux lines is suppressed by crystalline defects and impurities in the specimen, which are called pinning centers (PCs). Thus, artificially embedding crystalline defects as effective PCs is just a key strategy to improve the in-field performance of superconductors [1,3,7]. For the last fifteen years or so, doping of non-superconducting secondary phases such as $BaMO_3$ (M = Zr, Sn, Hf, etc.) and $RE_2O_3$ has been attempted to form those into effective PCs in REBCO thin films [8–12].

The flux pinning effect depends on the shape (dimensionality), orientation, size, and distribution of PCs. In particular, the dimensionality of PCs significantly affects the feature of flux pinning, as shown in Figure 1. For example, one-dimensional PCs such as columnar defects (CDs) exhibit a preferential direction for the flux pinning: the strong flux pinning occurs in the magnetic field direction along their long axis. Three-dimensional PCs such as nano-particles, on the other hand, have the morphology with no correlated orientation for flux pinning, resulting in the isotropic pinning force against any direction of magnetic field. These features of PCs play an important role in the modification of the $J_c$ properties in REBCO films: those parameters of PCs such as their shape and size, should be designed to meet the requirements for each application.

Swift-heavy-ion irradiation to high-$T_c$ superconductors produces amorphous CDs of damaged material parallel to the projectile direction through the electron excitation process rather than the nuclear collision process. The CDs produced by the irradiation effectively work as one-dimensional PCs [13–15]. The orientation of one-dimensional PCs determines the preferential direction of flux pinning [13,16]. Therefore, heavy-ion irradiation can be expected to modify the anisotropy of $J_c$ in high-$T_c$ superconductors by tuning the irradiation direction. In addition, the size and shape of CDs strongly depends on the electronic stopping power $S_e$, which is defined as energy loss of the incident ion per unit length via electronic excitation in the target material [17]: continuous CDs with thick diameter are formed at higher $S_e$ than a certain value and discontinuous CDs with thin diameter are located at intervals along the ion path at lower $S_e$ [18–20]. In particular, discontinuous CDs may provide more effective flux pinning in a wide magnetic field angular range, because the ends of discontinuous CDs can act as PCs even in magnetic field directions tilted from their long axis [21,22]. Thus, the discontinuity of CDs is also one of the important factors for the modification of the $J_c$ anisotropy in high-$T_c$ superconductors, as well as the direction-dispersion of CDs.

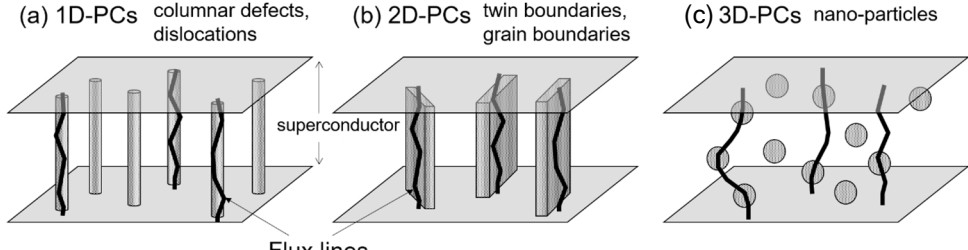

**Figure 1.** Sketch of the different dimensional categories for PCs: (**a**) 1D columnar (linear) defects, (**b**) 2D planar defects such as twin boundaries, and (**c**) 3D nano-particles.

A major advantage of using heavy-ion irradiation for the formation of CDs is that any CD configuration can be prepared by tuning the irradiation energy and the incident direction [23,24], independently from a fabrication process of samples (see Figure 2): the pinning structure can be efficiently designed to meet the requirements for different applications, which would be valuable for the development of high-performance coated conductors. In addition, unique pinning structures architected by the irradiations may enable us to find new physics of flux line dynamics. Therefore, heavy-ion irradiation to high-$T_c$ superconductors can provide the design criteria for the supreme pinning landscape making the most of the potential for flux pinning, which leads to $J_c$ close to the theoretical limit of critical current density, i.e., the pair-breaking critical current density.

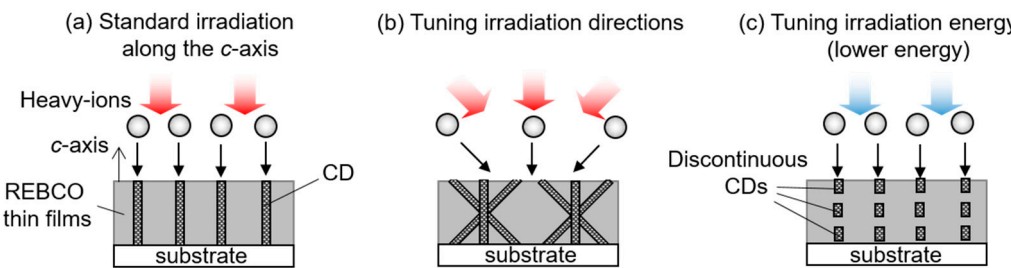

**Figure 2.** Schematic illustration of various configurations of CDs designed by heavy-ion irradiation: (**a**) continuous CDs parallel to the c-axis produced by standard irradiation, (**b**) direction-dispersed CDs installed by tuning the irradiation directions, (**c**) discontinuous CDs formed by adjusting irradiation energy to lower value.

In this paper, we describe the results of the modification of the $J_c$ properties in REBCO thin films and coated conductors, which were obtained by our studies through heavy-ion irradiation under various irradiation conditions. Most of previous works of other researchers using heavy-ion irradiation have focused on the improvement of $J_c$ at $B \parallel c$ where $J_c$ usually shows the minimum [13–15,18,19]. On the other hand, heavy-ion irradiation effects over a wide magnetic field angular range have not been well studied so far. By contrast, we focus especially on modification of the $J_c$ anisotropy in high-$T_c$ superconductors by using heavy-ion irradiation: our aim in this review is to improve $J_c$ in all magnetic field angular range from $B \parallel c$ to $B \parallel ab$ by using CDs and to explore breakthroughs for strong and isotropic pinning landscape in REBCO coated conductors. To meet the aim in this paper, we selected Xe ions as the irradiation ion species: the Xe-ion irradiation to REBCO thin films can provide large increase of $J_c$ without heavily damaging crystallinity even at a large amount of doses, $5.0 \times 10^{11}$ ions/cm$^2$ [24] and easily enables us to tune the morphology of CDs through the adjustment of the irradiation energy at a tandem accelerator of Japan Atomic Energy Agency (JAEA) used in our works. Firstly, we present the reduction of the $J_c$ anisotropy by using the direction-dispersed CDs, which are introduced by controlling the irradiation direction. Secondly, we report the influence of CDs tilted at small angle(s) relative to the *ab*-plane on the $J_c$ properties near $B \parallel ab$, which is one of key factors to improve $J_c$ in all magnetic field directions. In particular, we show the influence of CDs along the *ab*-plane on $J_c$ at $B \parallel ab$ by preparing an in-plane aligned *a*-axis-oriented YBCO film. Finally, we clarify the potential of discontinuous CDs for flux pinning in comparison with continuous CDs, where the morphology of CDs is controlled by the irradiation energy.

## 2. Experimental

The samples used in our works were mostly *c*-axis oriented YBCO thin films and GdBCO coated conductors. The *c*-axis oriented YBCO thin films were fabricated by a pulsed laser deposition (PLD) technique on (100) surface of SrTiO$_3$ single crystal substrates. The thickness of the films was about 300 nm. The GdBCO coated conductor, on the other hand, was fabricated on an ion-beam-assisted deposition (IBAD) substrate by a PLD method (Fujikura Ltd., Tokyo, Japan). The thickness of GdBCO layer is 2.2 μm and the self-field critical current $I_c$ of this tape with 5 mm width is about 280 A. The samples were cut from the tape of the GdBCO coated conductor. The Ag stabilizer layer on the superconducting layer was removed by a chemical process. The YBCO thin films and the samples cut from the GdBCO coated conductor were patterned into a shape of about 40 μm wide and 1 mm long micro-bridge before the irradiation.

The heavy-ion irradiations with Xe ions were performed using the tandem accelerator of JAEA in Tokai, Japan. Tuning of the discontinuity of CDs along the *c*-axis can be controlled by the irradiation energy. The values of $S_e$ for the Xe-ion irradiation energies above 200 MeV are above 2.9 keV/Å, which is above the threshold value of $S_e = 20$ keV/nm to create continuous CDs along the *c*-axis over the whole sample thickness for YBCO [17].

Thus, the irradiation with 200 MeV Xe ions was performed to install continuous CDs into YBCO thin films. In addition, the Xe-ion irradiation with 270 MeV was applied in order to create continuous CDs for GdBCO coated conductors, where the projectile length was longer than the thickness of 2.2 μm. Discontinuous CDs, on the other hand, were formed into YBCO thin films and GdBCO coated conductors by the irradiation with 80 MeV Xe ions, where the value of $S_e$ is below 20 keV/nm: the radius of CDs strongly fluctuates along the ion path and CDs are shortly segmented at intervals in their longitudinal direction when the $S_e$ is lower than the threshold value, as shown in Figure 3 [19,20,25]. All of the irradiation energies used in our works are enough for the projectile ranges to exceed the thickness of the samples: the incident ions pass through the superconducting layer completely.

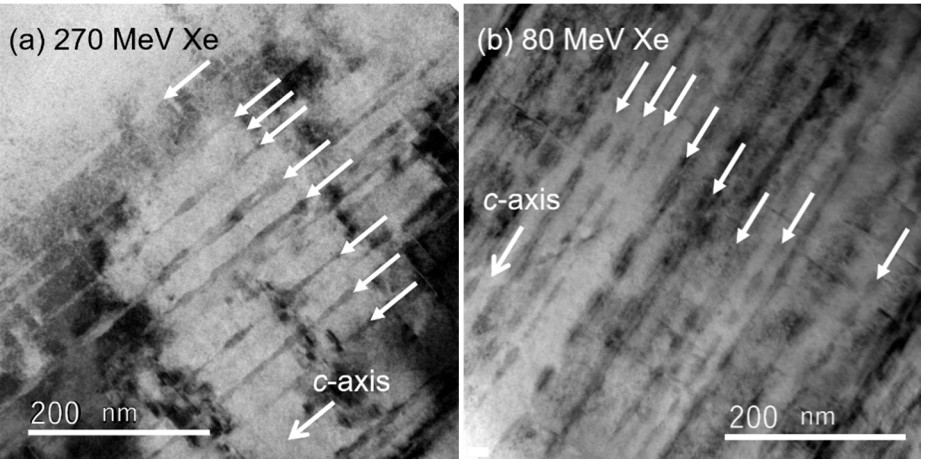

**Figure 3.** Cross-sectional TEM images of GdBCO coated conductors irradiated with (**a**) 270 MeV and (**b**) 80 MeV Xe ions, respectively. The irradiation dose is $1.94 \times 10^{11}$ ions/cm². The arrows indicate several ion tracks. Reprinted with permission from [25], copyright 2015 by IEEE.

The direction of CDs was adjusted by controlling the incident ion beam direction tilted off the *c*-axis by $\theta_i$, which was always directed perpendicular to the bridge direction of the sample (see Figure 4). When the irradiation directions are dispersed, the fluence in each irradiation direction is calculated by dividing the total fluence by the number of the irradiation directions. The fluence of the irradiation is often represented as a matching field $B_\varphi$: $B_\varphi$ is the magnetic field where the density of flux lines is equal to that of CDs, e.g., the fluence of $4.84 \times 10^{10}$ ions/cm² corresponds to $B_\varphi = 1$ T.

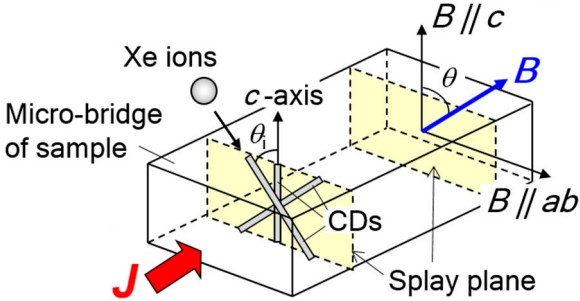

**Figure 4.** Sketch of the experimental arrangement in this work.

It should be noted that the introduction of irradiation defects causes a lattice distortion of the host matrix, which affects the superconducting properties such as critical temperature ($T_c$). The strain induces the oxygen vacancies [26], resulting in the reduction of $T_c$: the value of $T_c$ decreases when the fluence of the irradiation increases [24]. The strain also affects the $J_c$ properties through the influence on $T_c$: $J_c$ decreases largely, when the influence

of the strain increases excessively. Therefore, the irradiation fluences were adjusted to avoid heavy damage to the crystallinity in our works.

The cross sections of the irradiated samples were observed by conventional transmission electron microscopy (TEM) with a JEM-2000 EX instrument (JEOL, Tokyo, Japan) operating at 200 kV. The thin TEM specimens were prepared by a focused ion beam method using an Quanta 3D system (FEI, Hillsboro, Oregon, USA). The $J_c$ properties were measured through the transport properties by using a four-probe method. The $J_c$ was defined by a criterion of electric field, 1 μV/cm. The transport current was always perpendicular to the magnetic field and the *c*-axis (maximum Lorentz force configuration). The magnetic field angular dependences of $J_c$ were evaluated as a function of the angle $\theta$ between the magnetic field and the *c*-axis of the samples (see Figure 4).

## 3. Results and Discussion

### 3.1. Modification of $J_c$ Around B || c by Controlling Heavy-Ion Irradiation Angles

Heavy-ion irradiation can introduce CDs in any direction in a controlled manner, so we can install CDs at the magnetic field angles where the $J_c$ shows a minimum, one by one: the material processing with heavy-ions is one of effective ways to modify the $J_c$ anisotropy in high-$T_c$ superconductors, which enables us to push up overall $J_c$, as shown in Figure 5.

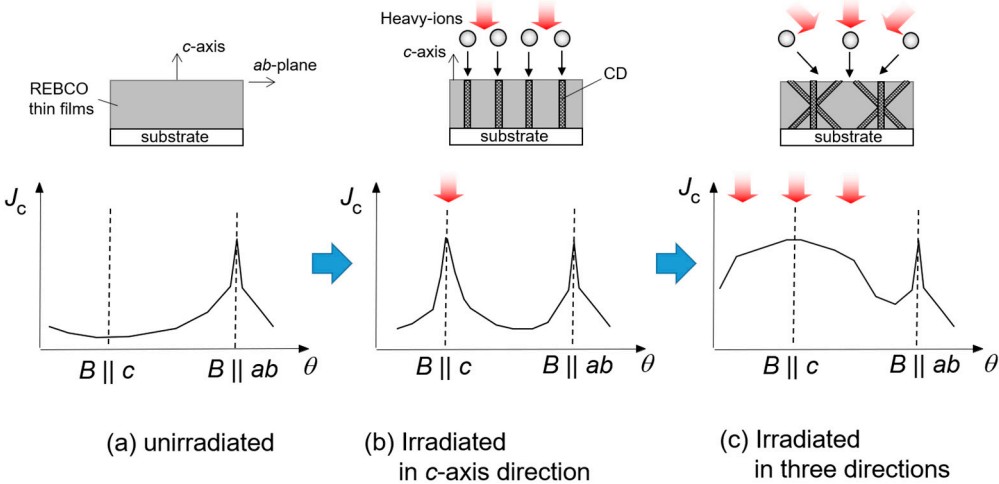

**Figure 5.** Schematic image of modification of the $J_c$ anisotropy by controlling the irradiation directions ((**a**) typical $J_c$ anisotropy of unirradiated high-$T_c$ superconductors, (**b**) modified $J_c$ anisotropy with CDs along the *c*-axis, (**c**) modified $J_c$ anisotropy with direction-dispersed CDs).

We first examined the influence of bimodal angular distribution of CDs consisting of CDs crossing at $\pm\theta_i$ relative to the *c*-axis on the $J_c$ properties in a wide magnetic field angular range [27,28]. Figure 6 shows the magnetic-field angular dependence of $J_c$ normalized by the self-magnetic-field critical current density $J_{c0}$ for YBCO thin films with the crossed CDs, which were installed by 200 MeV Xe ion irradiation with $B_\varphi$ = 2 T (c10-2: $\theta_i$ = $\pm 10°$, c25-2: $\theta_i$ = $\pm 25°$, c45-2: $\theta_i$ = $\pm 45°$, p06-2: parallel CD configuration of $\theta_i$ = 6°, and Pure: unirradiated samples). The magnetic field was rotated in the splay plane where the two parallel CD families are crossing each other, as shown in Figure 4. All the irradiated samples show an additional peak of the normalized $J_c$ around $B \parallel c$ ($\theta$ = 0°) for lower magnetic fields: the values of the normalized $J_c$ are enhanced around $B \parallel c$ compared to the unirradiated one. This indicates that CDs with any crossing angle work as effective PCs, pushing up the $J_c$ around $B \parallel c$. The influence of the crossing angle of CDs is evident in the shape of the additional peak around $B \parallel c$: the width of the normalized $J_c$ peak becomes broader when the crossing angle is larger. Therefore, the bimodal angular distribution of CDs can expand the magnetic field angular range where the normalized $J_c$ increases, by controlling the crossing angle.

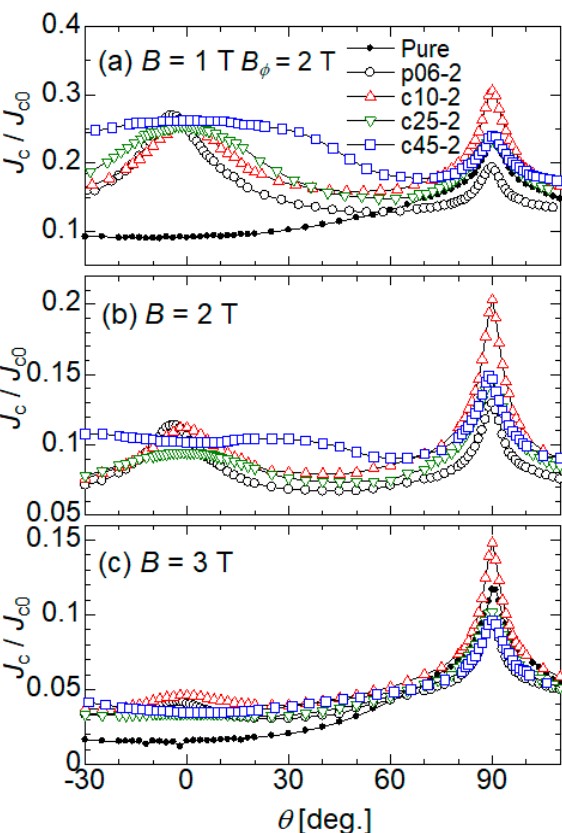

**Figure 6.** Magnetic field angular dependence of $J_c$ normalized by the self-magnetic-field critical current density $J_{c0}$ for YBCO thin films with the crossed CDs (c10-2: $\theta_i = \pm10°$, c25-2: $\theta_i = \pm25°$, c45-2: $\theta_i = \pm45°$, p06-2: parallel CD configuration of $\theta_i = 6°$, and Pure: unirradiated samples). Reprinted with permission from [28], copyright 2016 by IOP.

It is noteworthy that the crossover phenomenon from the broad-plateau-like behavior to the double peak emerges on the normalized $J_c$ around $B \parallel c$ for c45-2 when the magnetic field increases across the matching field of $B_\varphi = 2$ T: the normalized $J_c$ more rapidly reduces at $B \parallel c$ with increasing magnetic field, which results in a dip structure at $B \parallel c$ for c45-2 at 2 T, as shown in Figure 6. In general, the $J_c$ peak in the magnetic field angular dependence of $J_c$ is a sign of long-axis correlated flux pinning of CDs. Their-long-axis correlated flux pinning is maintained up to higher magnetic fields [29,30]. For the crossing angle of $\theta_i = \pm45°$, by contrast, the influence of the long-axis correlated flux pinning is weakened at $B \parallel c$, since the directions of CDs are far from the $c$-axis direction. Thus, the dip behavior at $B \parallel c$ is a sign of disappearance of their-long axis correlated flux pinning at $B \parallel c$.

The effective magnetic field angular region for flux pinning of CDs is described by a trapping angle $\varphi_t$, at which flux lines begin to be partially trapped by CDs [4]. The general formula of $\varphi_t$ is expressed as:

$$\varphi_t = \sqrt{2\,\varepsilon_p/\varepsilon_l} \tag{1}$$

where $\varepsilon_p$ is the pinning energy of CDs and $\varepsilon_l$ is the line tension of flux lines. The line tension energy of flux lines in anisotropic superconductors is given by the following equation:

$$\varepsilon_l(\Theta) \propto \varepsilon_0/\gamma^2\varepsilon(\Theta)^3 \tag{2}$$

where $\Theta$ is the angle between the magnetic field and the *ab*-plane, $\varepsilon_0$ is a basic energy scale, $\gamma$ is the mass anisotropy, and $\varepsilon(\Theta) = (\sin^2\Theta + \gamma^{-2}\cos^2\Theta)^{1/2}$ [4]. The trapping angle $\varphi_t$ is experimentally estimated as the difference in the angle between the peak value and the minimum one on the magnetic field angular dependence of $J_c$ [31]. For p06-2, the

value of $\varphi_t$ is ~55° at $B < B_\varphi$, which is estimated from Figure 6a. Using this value of $\varphi_t$ as the trapping angle of CDs parallel to the $c$-axis approximately and $\gamma = 5$ together with equations (1) and (2), the value of $\varphi_t$ for CDs tilted at $\theta_i = 45°$ is about 37°. Therefore, CDs tilted at $\theta_i = 45°$ hardly contribute to trapping flux lines at $B \parallel c$: CDs tilted at $\theta_i = 45°$ does not work as their-long-axis correlated PCs for $B \parallel c$.

The bimodal angular distribution of CDs for $\theta_i \pm 45°$ gives rise to the drop in $J_c$ at the mid-direction of the crossing angle. Secondly, we investigated the flux pinning properties for a trimodal angular distribution of CDs consisting of CDs crossing at $\theta_i = 0°$ and $\pm 45°$ (referred to as the "standard" trimodal-configuration), in order to obtain high $J_c$ with no drop over a wide magnetic field angular region [32]. In addition, another geometry for the trimodal configuration was prepared, where a splay plane defined by the three irradiation angles is parallel to the transport current direction (referred to as "another" trimodal-configuration), as shown in Figure 7: the two trimodal configurations enable us to elucidate the influence of the splay plane direction on the $J_c$ properties directly. Figure 8 shows the magnetic field angular dependence of normalized $J_c$ by $J_{c0}$ ($= j_c$) at several magnetic fields from 1 T up to 5 T for YBCO thin films with the trimodal angular configurations of CDs. A large enhancement of $j_c$ centered at $B \parallel c$ can be seen for all the irradiated samples. In particular, both the trimodal angular configurations show a much broader peak with larger $j_c$ than that of the parallel CD configuration. It should be noted that there is no drop of $j_c$ at $B \parallel c$ for both the trimodal configurations. This result indicates that the three parallel CD families tilted at $\theta_i = 0°$ and $\pm 45°$ effectively work as strong PCs in each irradiation direction: flux pinning at $B \parallel c$ where CDs tilted at $\theta_i = \pm 45°$ slightly contribute to trapping flux lines, is reinforced by CDs along the $c$-axis.

### "Another" trimodal-configuration

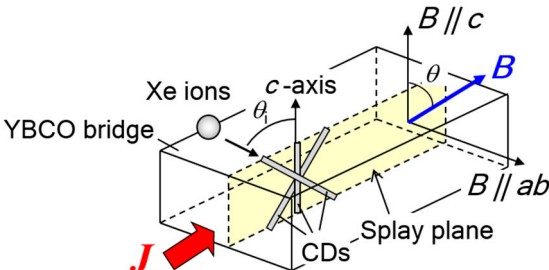

**Figure 7.** Sketch of CDs dispersed in geometry of "another" trimodal-configuration, where a splay plane defined by the three irradiation angles is parallel to the transport current direction. Reprinted with permission from [32], copyright 2016 by IOP.

Interestingly, the behaviour of $j_c$ around $B \parallel c$ strongly depends on the direction of the splay plane for the trimodal configuration of CDs: the $j_c$ of another trimodal-configuration shows a peak at $B \parallel c$, whereas standard one exhibits not so much a peak as a plateau-shaped curve. In addition, the height of $j_c$ peak for another tirmodal-configuration is higher than the value of $j_c$ at $B \parallel c$ for standard one. For standard trimodal-configuration, sliding motion of flux lines occurs along the tilted CDs at $B \parallel c$ because of the splay plane parallel to the Lorentz force, resulting in the reduction of the pinning efficiency [33]. The crossed CDs for another trimodal-configuration, by contrast, suppress the motion of flux lines efficiently, since flux lines move across the crossed CDs by the Lorentz force. Thus, the splay plane parallel to the transport current direction provides stronger flux pinning at $B \parallel c$, like planar PCs. Furthermore, the $j_c$ of another trimodal-configuration is the highest even when the magnetic field is tilted from the c-axis. This is probably due to the entanglement of flux lines induced in a mesh of the splay plane tilted from the magnetic field, where the motion of flux lines is suppressed [34]. These results suggest that the direction of the splay plane is one of key factors for flux pinning of direction-dispersed CDs, as well as the degree of the direction-dispersion [32,35].

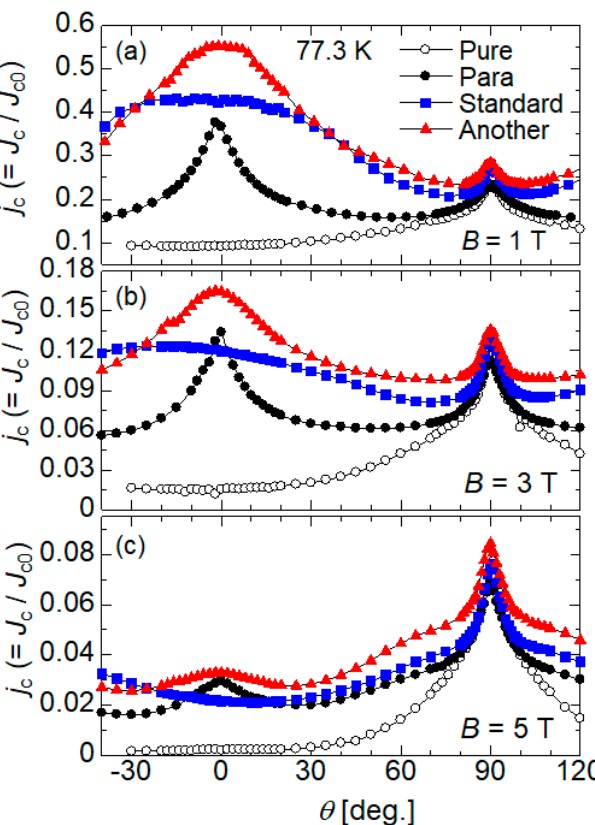

**Figure 8.** Magnetic-field angular dependence of $J_c$ normalized by the self-magnetic-field critical current density $J_{c0}$ for YBCO thin films with various CD configurations (Pure: unirradiated samples, Para: parallel CD configuration of $\theta_i = 0°$, Standard: standard trimodal-configuration, and Another: another trimodal-configuration). Reprinted with permission from [32], copyright 2016 by IOP.

### 3.2. Modification of $J_c$ Anisotropy by Controlling Number of Heavy-Ion Irradiation Directions

We further increased the number of the directions of CDs by controlling the irradiation directions (see Figure 9), in order to spread the strong pinning effect of CDs over a wider magnetic field angular range. Figure 10 shows the magnetic field angular dependence of $J_c$ and $n$-values for YBCO thin films with direction-dispersed CDs, where the number of directions of CDs was applied from one to five every 30 degrees [36]. The $n$-value is estimated from a linear fit to empirical formula of electric field ($E$) versus current density ($J$), $E \sim J^n$ in the range of 1 to 10 µV/cm. The $n$-value is equivalent to $U_0 / k_B T$ ($U_0$: pinning potential energy) [37,38], representing thermal activation for flux motion. When the number of CD directions is increased, the angular region with high $J_c$ is more expanded. Note that the height of the $J_c$ peak at $B \parallel c$ declines, since the density of CDs decreases with increasing number of irradiation directions in this work. Thus, the large direction-dispersion of CDs is effective for the enhancement of $J_c$ over a wider magnetic field angular region centered at $B \parallel c$.

The $J_c$ around $B \parallel ab$, on the other hand, does not seems to be affected by flux pinning of the direction-dispersed CDs: both $J_c$ and $n$-value at $\theta = 90°$ rather tend to decrease with increasing the degree of the direction-dispersion of CDs. One of the reasons for the reduction of $J_c$ at $B \parallel ab$ by the introduction of CDs is the damage on the superconductivity and/or the $ab$-plane-correlated PCs [16]. It should be noted that the sample of Quintmodal contains CDs crossing at $\pm 30°$ relative to the $ab$-plane (i.e., $\theta_i = \pm 60°$); nevertheless, the crossed CDs do not seem to contribute to the pinning interaction around $B \parallel ab$. Figure 11 represents the magnetic field angular dependence of $J_c$ for YBCO thin films including bimodal angular configurations of CDs with $\theta_i = \pm 30°$ and $\pm 60°$ relative to the $c$-axis, respectively [39]. The crossing angle of $\pm 30°$ relative to the $c$-axis induces the enhancement



of $J_c$ over a wide angular region centered at $B \parallel c$. The crossing of CDs at $\pm 30°$ relative to the *ab* plane, i.e., $\theta_i = \pm 60°$, by contrast, is ineffective in pushing up the $J_c$ at the mid-direction of the crossing angle, i.e., at $B \parallel ab$, whereas the peak of $J_c$ emerges at $\theta = \pm 60°$. These results indicate that the flux pinning around $B \parallel ab$ is hardly affected even by CDs tilted toward the ab-plane, which significantly differs from the flux pinning of CDs at $B \parallel c$. Thus, the flux pinning of CDs around $B \parallel ab$ is a new issue for the complete reduction of the $J_c$ anisotropy.

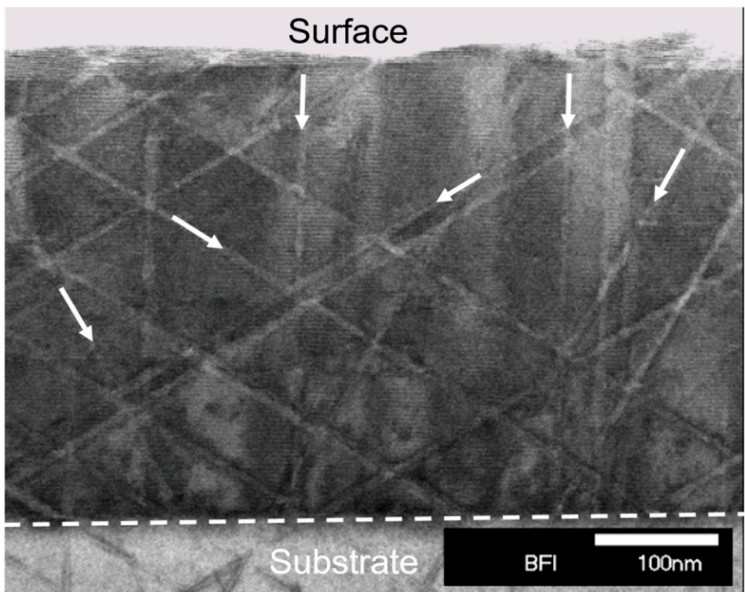

**Figure 9.** Bright field TEM image showing CDs tilted at $\theta_i = 0°$, $\pm 30°$ and $\pm 60°$ relative to the *c*-axis of the YBCO film, which are installed by 200 MeV Xe ion irradiation. The arrows indicate several ion tracks. Reprinted with permission from [36], copyright 2018 by IOP.

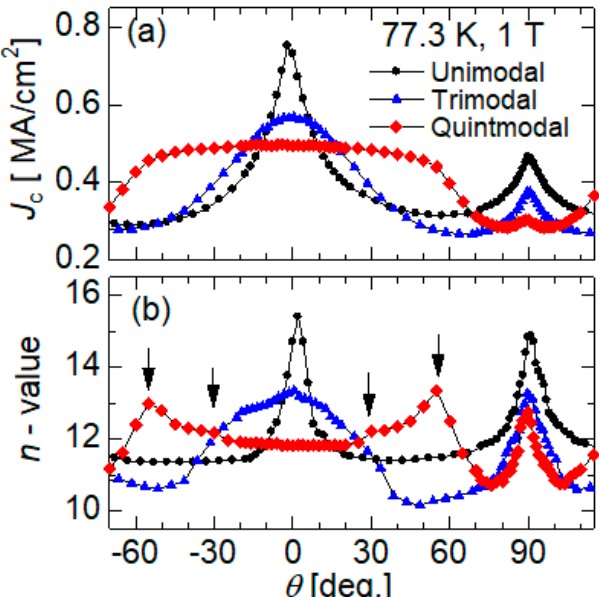

**Figure 10.** Magnetic-field angular dependence of $J_c$ (upper, (**a**)) and *n*-value (lower, (**b**)) for YBCO thin films with various CD configurations (Unimodal: parallel CD configuration with $\theta_i = 0°$, Trimodal: trimodal-configuration with $\theta_i = 0°$ and $\pm 30°$, and Quintmodal: quintmodal-configuration with $\theta_i = 0°$, $\pm 30°$ and $\pm 60°$). The arrows indicate the peaks or the shoulder on the $n(\theta)$ curve for Quintmodal. Reprinted with permission from [37], copyright 2013 by IEEE.

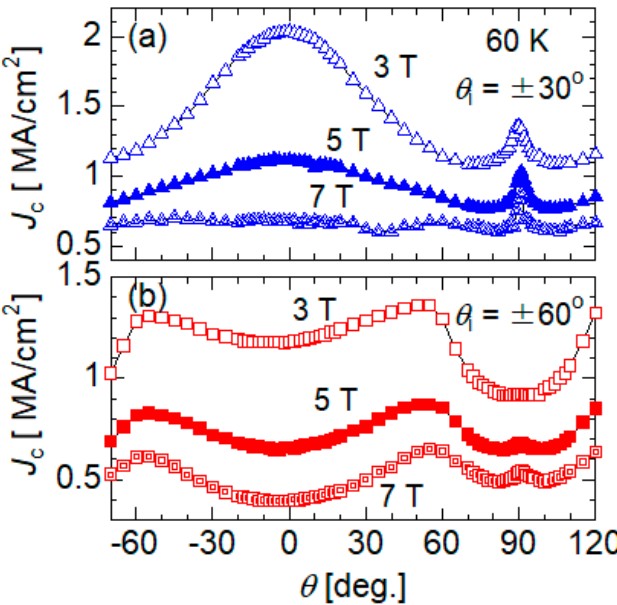

**Figure 11.** Magnetic-field angular dependence of $J_c$ at temperature of 60 K and magnetic field of 3 T to 7 T for YBCO thin films with CDs crossing at (**a**) $\theta_i = \pm30°$ and (**b**) $\pm60°$ relative to the *c*-axis, respectively. Reprinted with permission from [36], copyright 2018 by IOP.

### 3.3. Modification of $J_c$ Around $B \parallel ab$ by Controlling Heavy-Ion Irradiation Directions

A significant enhancement of $J_c$ at $B \parallel c$ has been caused by the introduction of artificially PCs, which is much higher than $J_c$ at $B \parallel ab$ now [40]. Thus, the improvement of $J_c$ at $B \parallel ab$ has been required at the next step, in order to increase overall $J_c$. The influence of CDs on the flux pinning at $B \parallel ab$, however, has not been well studied so far, because the $J_c$ at $B \parallel ab$ is the highest innately due to the electronic mass anisotropy in high-$T_c$ superconductors [4] and the introduction of CDs is generally difficult in the direction close to the *ab*-plane. In contrast, heavy-ion irradiation can be an effective tool even for exploring the flux pinning effect of CDs at $B \parallel ab$, because CDs can be installed in any direction by adjusting the irradiation direction.

GdBCO-coated conductors were irradiated with 270 MeV Xe-ions, where the irradiation angle $\Theta_i$ relative to the *ab*-plane was controlled in the range from $\pm5°$ to $\pm15°$ relative to the *ab*-plane in order to install crossed CDs around the *ab*-plane [41]. The cross-sectional TEM image of the GdBCO-coated conductor irradiated at $\Theta_i = \pm10°$, Figure 12, shows the formation of continuous CDs along the irradiation directions. At the bottom part of the GdBCO layer, by contrast, some CDs become thinner and indicate angular dispersion in the irradiation directions. This is due to smaller value of $S_e$ than the threshold value of 20 keV/nm for the formation of continuous CDs [17,42], because the $S_e$ changes from 29.1 to 7.40 keV/nm through the GdBCO layer for the oblique irradiation at $\Theta_i = 10°$.

Figure 13 shows the magnetic field angular dependence of $J_c$ for the irradiated samples with $\Theta_i = \pm5°$, $\pm10°$, and $\pm15°$, respectively. The CD crossing-angles of $\Theta_i \leq \pm15°$ significantly affect the magnetic field angular variation of $J_c$ around $B \parallel ab$. The introduction of crossed CD at $\Theta_i = \pm15°$ provides a triple peak of $J_c$ centered at $B \parallel ab$, where a large $J_c$ peak exists at $B \parallel ab$ and the other two $J_c$ peaks emerge around $\theta = 75°$ and 105°, independently each other. This behavior is in contrast to the case of CDs crossing at $\theta_i \leq \pm30°$ relative to the *c*-axis, which shows a single peak of $J_c$ centered at $B \parallel c$, as represented in Figures 6 and 11. As the crossing-angle of $\Theta_i$ decreases, the two divided peaks of $J_c$ at $\pm\Theta_i$ overlap with the central $J_c$ peak at $B \parallel ab$: a single peak centered at $\theta = 90°$ occurs for the crossing angles of $\Theta_i \leq \pm10°$. In particular, the crossing angle of $\Theta_i = \pm5°$ provides the large and sharp $J_c$ peak at $B \parallel ab$, showing the highest value of all the samples at $B \parallel ab$. To our knowledge, it is the first confirmation that CDs contribute to the improvement of $J_c$ at $B \parallel ab$.

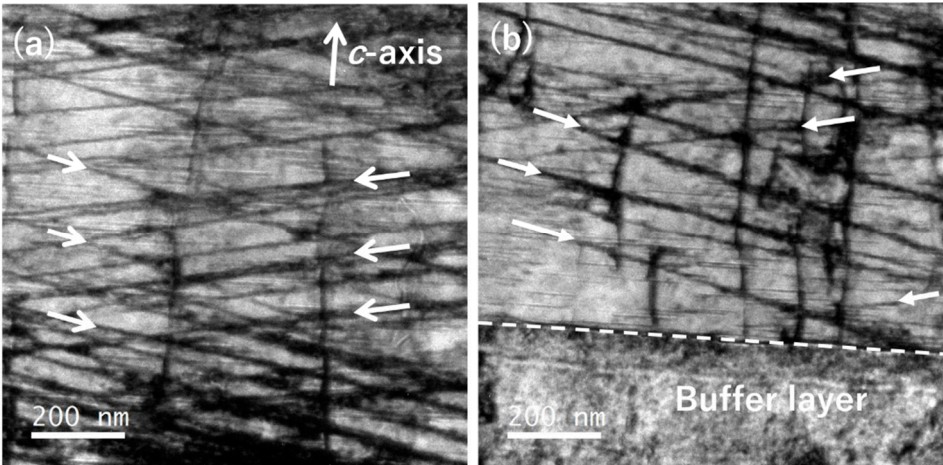

**Figure 12.** Cross-sectional TEM images of a GdBCO coated conductor irradiated with 270 MeV Xe ions at $\Theta_i = \pm10°$ relative to the *ab*-plane (**a**) near the surface and (**b**) at the bottom part of GdBCO layer, respectively. The arrows show several ion tracks. Reprinted with permission from [41], copyright 2017 by IEEE.

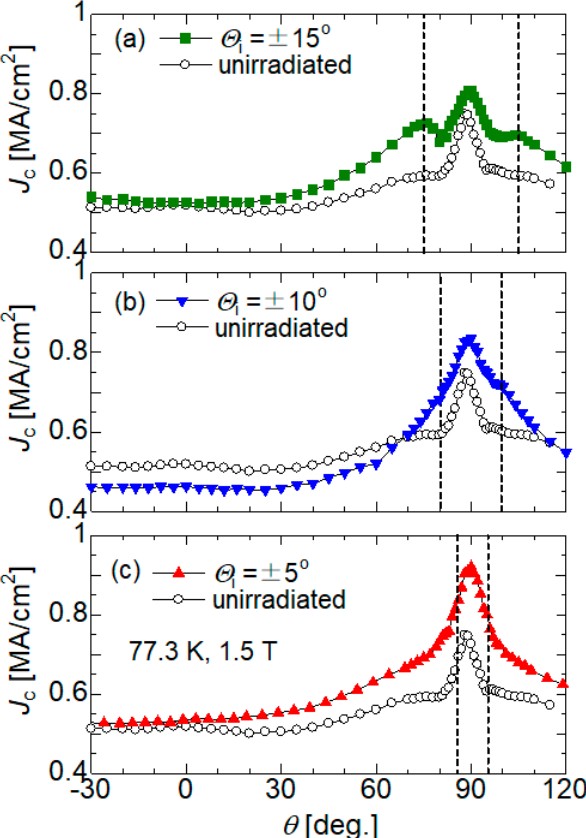

**Figure 13.** Magnetic-field angular dependence of $J_c$ at 77.3 K, 1.5 T in GdBCO coated conductors with crossed CDs at $\pm\Theta_i$ relative to the *ab*-plane ((**a**) $\Theta_i = \pm15°$, (**b**) $\Theta_i = \pm10°$, and (**c**) $\Theta_i = \pm5°$). The broken lines are drawn at the positions of the irradiation angles. Reprinted with permission from [41], copyright 2017 by IEEE.

These behaviors are closely associated with the elastic properties of flux lines around $B \parallel ab$. The line tension energy of flux lines becomes very strong at $B \parallel ab$, where the core of flux lines shows the elliptical nature in anisotropic superconductors. The strong line tension of flux lines significantly affects the trapping angle $\varphi_t$ of CDs tilted toward the

*ab*-plane. In general, the value of $\varphi_t$ for CDs along the *c*-axis (i.e., at $\Theta_i = 90°$) is about 65° in GdBCO coated conductors [25]. The $\varphi_t$ for CDs tilted at small angle of $\Theta_i$, on the other hand, can be evaluated by substituting the value of $\varphi_t \sim 65°$ for $\Theta_i = 90°$ and $\gamma = 5$ together with equations (1) and (2): $\varphi_t \sim 6.6°$ for $\Theta_i = 5°$, $\varphi_t \sim 8.7°$ for $\Theta_i = 10°$, and $\varphi_t \sim 11.9°$ for $\Theta_i = 15°$. Thus, the trapping angles of CDs tilted toward the *ab*-plane becomes very small: flux lines are hardly trapped along CDs when the magnetic field direction is displaced from the direction of CDs even slightly. In particular, the trapping angle for CDs tilted at $\Theta_i \geq 10°$ is smaller than the CD tilt-angle $\Theta_i$, suggesting that the tilted CDs hardly affect the flux pinning at $B \parallel ab$. Therefore, CDs tilted at $\Theta_i \geq 10°$ and the *ab*-plane correlated PCs provide flux pinning independently. The CDs tilted at $\Theta_i = 5°$, on the other hand, can fully contribute to the improvement of $J_c$ at $B \parallel ab$, because the trapping angle exceeds the value of $\Theta_i$.

### 3.4. Modification of $J_c$ at $B \parallel ab$ by Heavy-Ion Irradiation along the a-Axis

An in-plane aligned *a*-axis-oriented YBCO film offers an excellent opportunity for further exploration into the influence of CDs on the flux pinning at $B \parallel ab$, since we can easily install CDs along the *ab*-plane with the ion-beam normal to the film [43]. We prepared the in-plane aligned *a*-axis-oriented YBCO film by a PLD technique with an ArF excimer laser, where a (100) SrLaGaO$_4$ substrate with Gd$_2$CuO$_4$ buffer layer was used to promote the in-plane orientation of YBCO thin film [44]. The film was patterned into the shape of a microbridge so as to make the bridge direction parallel to the *b*-axis, where transport current can be applied along the *ab*-plane (see Figure 14). Both the in-plane-aligned texture of the film and the experimental arrangement enable us to remove the extra effect such as the interlayer Josephson current and the channel flow of flux lines along the CuO$_2$ plane, providing deeper insights on the nature of flux pinning of CDs along the *ab*-plane.

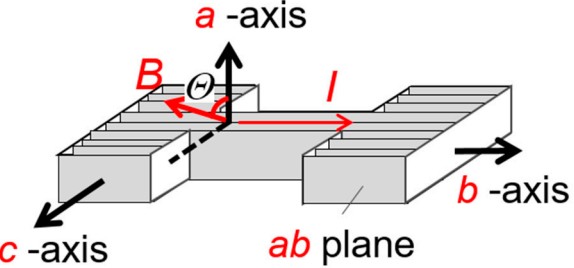

**Figure 14.** Sketch of the experimental arrangement using the in-plane aligned *a*-axis-oriented YBCO film. Reprinted with permission from [43], copyright 2019 by IEEE.

The in-plane aligned *a*-axis-oriented YBCO thin film showed good *a*-axis orientations without other orientations for the X-ray $\theta$-$2\theta$ diffraction pattern, as shown in Figure 15. In addition, X-ray diffraction $\varphi$ scanning using the (102) plane of the YBCO film before the irradiation indicated two-fold symmetry, since strong peaks stood out at around 90° and 270° in the inset of Figure 15. Therefore, the in-plane aligned *a*-axis-orientated microstructure can be confirmed on the film used in this work.

A cross-sectional TEM image of the in-plane aligned *a*-axis oriented YBCO film after the irradiation with 200 MeV Xe ions is shown in Figure 16a. The straight CDs along the *a*-axis are elongated through the thickness of the YBCO film. Figure 16b shows the plan-view TEM image of the *a*-axis oriented YBCO thin film after the irradiation. The CDs formed by the ion beam along the a-axis are roughly elliptical in shape, whereas CDs parallel to the *c*-axis are usually circular [17,45]. In general, the shape of CDs depends on the direction of the incident ions relative to the crystallographic axes in high-$T_c$ superconductors, because the anisotropy of thermal diffusivity causes more severe irradiation damage for the creation of CDs along the *a*- and/or the *b*-axis [17].

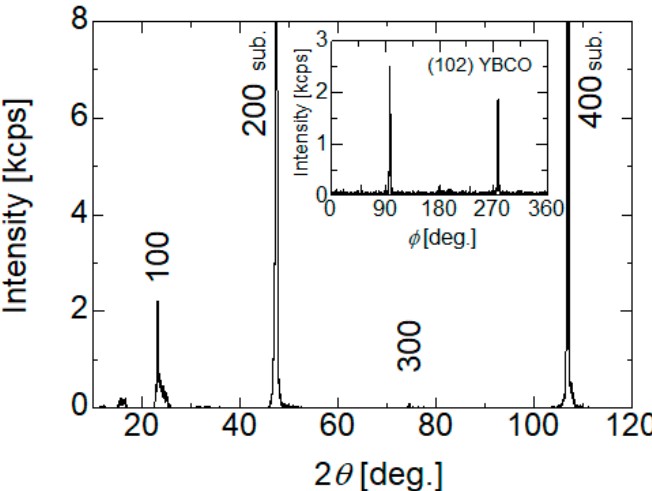

**Figure 15.** X-ray diffraction $\theta$-$2\theta$ scan of the in-plane aligned a-axis oriented YBCO thin film before the irradiation. Inset: X-ray $\varphi$ scan using (102) plane of the YBCO thin film before the irradiation. Reprinted with permission from [43], copyright 2019 by IEEE.

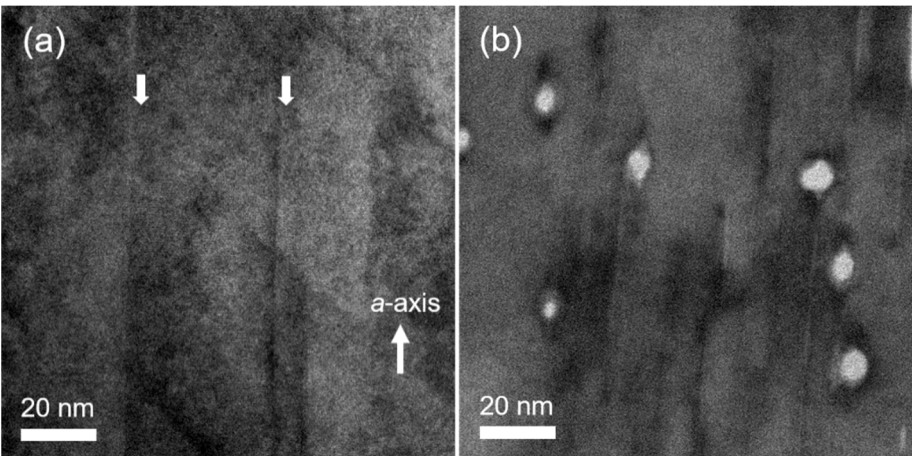

**Figure 16.** (**a**) Cross-sectional and (**b**) plan-view TEM images of the a-axis oriented YBCO thin film irradiated with 200 MeV Xe ions along the a-axis. The arrows indicate several ion tracks. Reprinted with permission from [43], copyright 2019 by IEEE.

Figure 17 represents the magnetic field dependence of $J_c$ at 72 K for the *a*-axis oriented YBCO film before and after the irradiation. The $J_c$ at $B \parallel c$ is reduced by the introduction of CDs along the *a*-axis, especially for high magnetic fields. The CDs along the *a*-axis hardly interact with flux lines at $B \parallel c$, since the CDs are perpendicular to the magnetic field direction. Moreover, CDs perpendicular to magnetic field direction create easy channel for flux lines to creep along the length of the CDs [46]. In addition to these deterioration effects, the irradiation damage to the host matrix causes the pronounced reduction of $J_c$ at $B \parallel c$.

The introduction of CDs along the *a*-axis, on the other hand, hardly reduces the absolute value of $J_c$ at $B \parallel a$, even though the $J_c$ is affected by the local irradiation damage to the $CuO_2$ planes as well as the $J_c$ at $B \parallel c$. It should be noted that the normalized $J_c$ by $J_{c0}$ increases after the irradiation, especially for high magnetic fields (see the inset of Figure 17). This behavior suggests that CDs contribute to the flux pinning at $B \parallel ab$. For low magnetic fields, by contrast, the pinning effect of CDs along the *a*-axis is hardly visible even on the normalized $J_c$. This is attributed to the presence of the naturally growth defects such as stacking faults in the film: Such pre-existing defects act as *ab*-plane correlated PCs both before and after the irradiation, which obscures the pinning effect of CDs, especially for low magnetic fields.

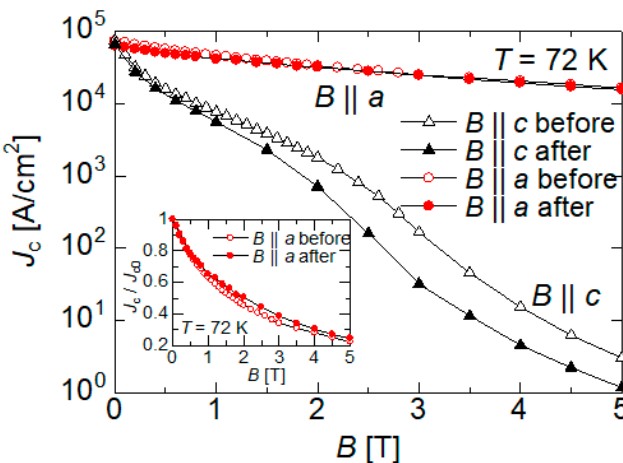

**Figure 17.** Magnetic field dependence of $J_c$ at $B \parallel c$ and at $B \parallel a$ in the $a$-axis oriented YBCO thin film before and after the irradiation. Inset: $J_c$ normalized by self-field critical current density $J_{c0}$ as a function of magnetic field along the $a$-axis. Reprinted with permission from [43], copyright 2019 by IEEE.

### 3.5. Modification of the $J_c$ Anisotropy by Controlling the Heavy-Ion Irradiation Energy

The modification of the $J_c$ anisotropy in high-$T_c$ superconductors is sensitive to direction-dispersions of CDs, as mentioned in the previous sections. Another way to modify the $J_c$ properties by CDs is to tune the morphologies of CDs. Especially for the morphology of short segmented (i.e., discontinuous) CDs, the ends of the discontinuous CDs can provide a variety of additional pinning effects: the ends of the segmented CDs can trap flux lines in magnetic field tilted from their long axis [21,22] and the existence of gaps in the segmented CDs can suppress thermal motion of flux lines, as shown in Figure 18. Furthermore, the volume fraction of CDs relative to the superconducting area can be minimized for discontinuous CDs, since CDs are shortly segmented: the reduction of the volume fraction of the crystalline defects suppresses the degradation of the superconductivity associated with the introduction of PCs, leading to the improvement of the absolute value of $J_c$ in a whole magnetic field angular region [19,20]. For iron-based superconductors, the morphology of CDs formed by heavy-ion irradiation tends to be discontinuous, which induces the remarkable improvement of $J_c$ [47–49]. The morphology of CDs in high-$T_c$ superconductors can be tuned by adjusting the irradiation energy for heavy-ion irradiation. In addition, the pinning effect of discontinuous CDs can be compared directly with that of continuous ones under same irradiation conditions except for the irradiation energy: the heavy-ion irradiations with different irradiation energies enable us to clarify the superiority of discontinuous CDs in the flux pinning effect over continuous CDs.

We first compared the flux pinning properties of discontinuous CDs with those of continuous ones when their long axis is parallel to the $c$-axis: GdBCO-coated conductors were irradiated with 80 MeV and 270 MeV Xe-ions along the $c$-axis, respectively [25]. For the irradiated sample with 270 MeV Xe ions, the straight and continuous CDs with the diameter of 4-11 nm penetrate the superconducting layer along the $c$-axis, as shown in Figure 3a. The value of $S_e$ calculated using SRIM code varies from 3.0 to 2.8 keV/Å through the superconducting layer with the thickness of 2.2 μm for the 270 MeV Xe-ion irradiation, so that the continuous CDs are formed over the whole sample. The 80 MeV Xe-ion irradiation, by contrast, produces short segmented CDs in their longitudinal direction along the $c$-axis, as shown in Figure 3b: the length of the segmented CDs with the diameter of 5-10 nm varies from 15 to 50 nm along their length, while the gaps between the segmented CDs is also variable, ranging between 15 and35 nm. The formation of discontinuous CDs is attributed to the value of $S_e$ changing from 2.0 to 1.4 keV/Å for the 80 MeV Xe ions into REBCO thin films [17,19,20].

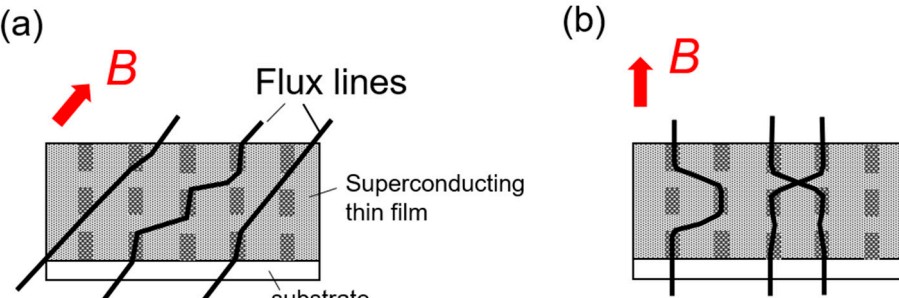

**Figure 18.** Schematic images of flux pinning peculiar to discontinuous CDs. (**a**) Ends of the discontinuous CDs work as PCs in magnetic field tilted off their long axis, which provide effective flux pinning over a wide magnetic field angular range. (**b**) Existence of gaps in the discontinuous CDs can suppress thermal motion of kinks of flux lines, which further improve $J_c$ in comparison with continuous CDs.

Figure 19 shows the magnetic field angular dependences of $J_c$ at 70 K and 84 K in GdBCO-coated conductors irradiated with 80 MeV and 270 MeV Xe ions, respectively. The 80 MeV irradiation causes higher $J_c$ in all magnetic field directions compared to the 270 MeV irradiation, which becomes more pronounced at lower temperature of 70 K. The high $J_c$ at $B \parallel c$ for the 80 MeV irradiation is attributed to the existence of gaps in the segmented CDs, which induce the suppression of thermal motion of flux lines (see Figure 18b). In addition, the ends of discontinuous CDs can trap flux lines in magnetic field tilted from their long axis, as shown in Figure 18a. These flux pinning effects of discontinuous CDs become more remarkable at lower temperature where a core size of flux line approaches the thin diameter of the discontinuous CDs. Moreover, discontinuous CDs more minimize the degradation of the superconductivity associated with the introduction of PCs compared with continuous CDs. Thus, the discontinuity of CDs can contribute to further enhancement of $J_c$.

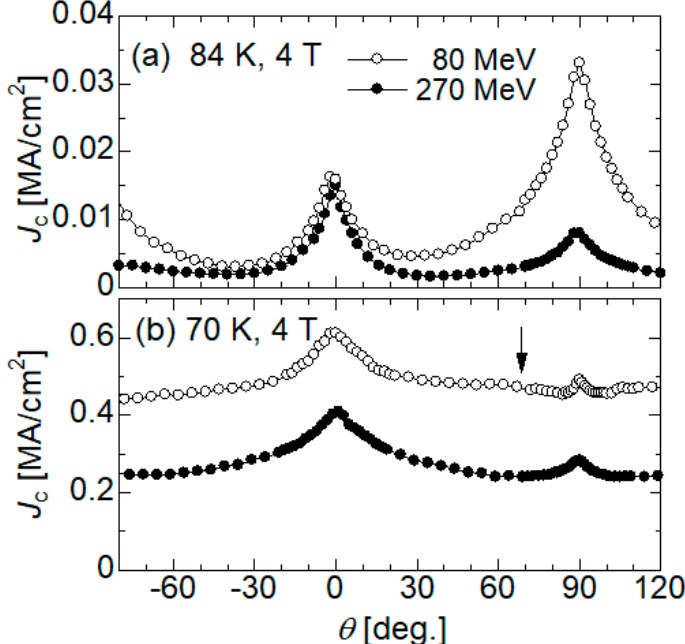

**Figure 19.** Magnetic field angular dependence of $J_c$ at 4 T for the irradiated samples with 80 MeV and 270 MeV Xe ions ((**a**) 84 K and (**b**) 70K). Reprinted with permission from [25], copyright 2015 by IEEE.

The superior flux pinning effect of discontinuous CDs can be further modified by tuning the direction-dispersion. We irradiated GdBCO coated conductors with 80 MeV Xe ions, where the incident ion beams were tilted from the *c*-axis by $\theta_i$ to introduce various kinds of direction-dispersed CDs: a parallel configuration composed of CDs parallel to the *c*-axis, bimodal angular configuration composed of CDs tilted at $\theta_i = \pm 45°$ relative to the *c*-axis, and trimodal angular configuration composed of CDs tilted at $\theta_i = 0°$ and $\pm 45°$ [50].

Figure 20a shows a cross-sectional TEM image of the GdBaCuO-coated conductor irradiated with 80 MeV Xe-ions at $\theta_i = 0°$ and $\pm 45°$. The morphologies of CDs are schematically emphasized in Figure 20b. Interestingly, the 80 Me V Xe-ion beams create CDs with different morphologies depending on the irradiation angles of $\theta_i$: thick and elongated CDs are formed along the ion path at $\theta_i = 45°$, whereas the 80 MeV ions at $\theta_i = 0°$ creates short segmented CDs along their length. In general, the morphology of CDs is determined by the value of $S_e$, which is the energy transferred from the incident ions for the electronic excitation. A thermal spike model [51,52], which is one of models to interpret the formation of irradiation defects through the electronic excitation, can describes the direction-dependent morphologies of CDs in high-$T_c$ superconductors by considering the anisotropy of thermal diffusivity [17,50]. According to the thermal spike model, the energy of the electronic excitation is converted into the thermal energy of lattice, which is the source for the formation of irradiation defects. In high-$T_c$ superconductors, the thermal diffusivity along the *c*-axis is smaller than that along other crystallographic axes, which results in the suppression of a temperature spread in the planes containing the *c*-axis. Thus, the incident ion beam tilted from the *c*-axis causes more severe structural damage, resulting in the formation of elongated CDs with a thicker diameter.

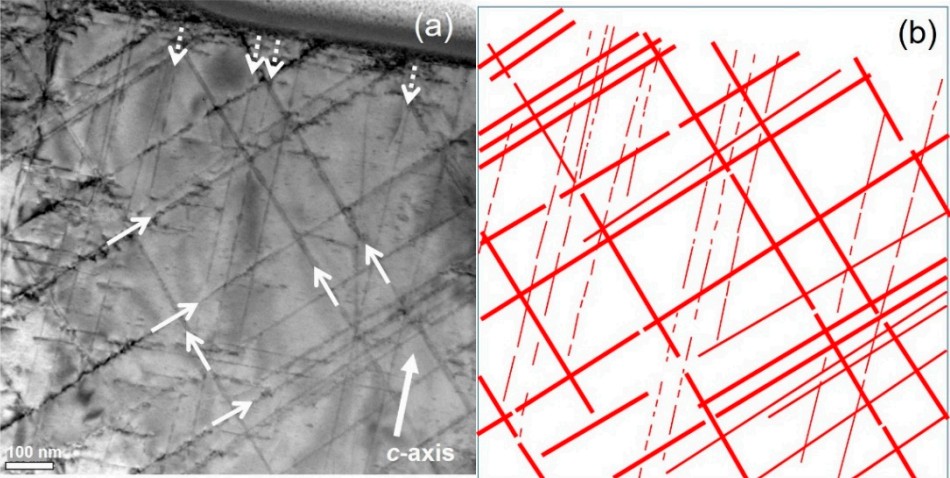

**Figure 20.** (**a**) Cross-sectional TEM image of the GdBCO coated conductor irradiated with 80 MeV Xe ions at $\theta_i = 0°$ and $\pm 45°$. Several continuous CDs are indicated by the solid arrows and discontinuous CDs along the *c*-axis are indicated by the dotted arrows. (**b**) Schematic image emphasizing the morphologies of CDs in the cross-sectional TEM image. Reprinted with permission from [50], copyright 2020 by the Japan Society of Applied Physics.

Figure 21 shows the magnetic field angular dependence of $J_c$ for GdBCO coated conductors irradiated with 80 MeV and 270 MeV Xe ions, where the irradiation angles are $\theta_i = 0°$ for the parallel CD configurations and $\theta_i = 0°$, $\pm 45°$ for the trimodal angular configuration, respectively. The trimodal angular distribution shows higher $J_c$ values than the parallel CD configuration at 70 K under the same irradiation energy. This suggests that the direction-dispersion of CDs is more effective to enhance the flux pinning over a wide magnetic field angular region, as mentioned in Section 3.2. It is noteworthy that the trimodal angular configuration produced by 80 MeV Xe ions shows the highest $J_c$ in all the CD configurations over the whole magnetic field angular region at 70 K. The 80 MeV trimodal configuration consists of short segmented CDs along the *c*-axis and elongated CDs

crossing at $\theta_i = \pm 45°$, as shown in Figure 20a. For $B \| c$, the motion of double kinks of flux lines peculiar to one-dimensional PCs is suppressed by the gaps between the segmented CDs, as shown in Figure 18b. Furthermore, continuous CDs crossing at $\theta_i = \pm 45°$ assist in trapping the unpinned segments of flux lines, as shown in Figure 22a. The pinning of kinks of flux lines is effective for further improvement of $J_c$ [53,54]. Therefore, the combination of discontinuous CDs and continuous ones crossing at $\theta_i = \pm 45°$ provides the enhancement of $J_c$ at $B \| c$.

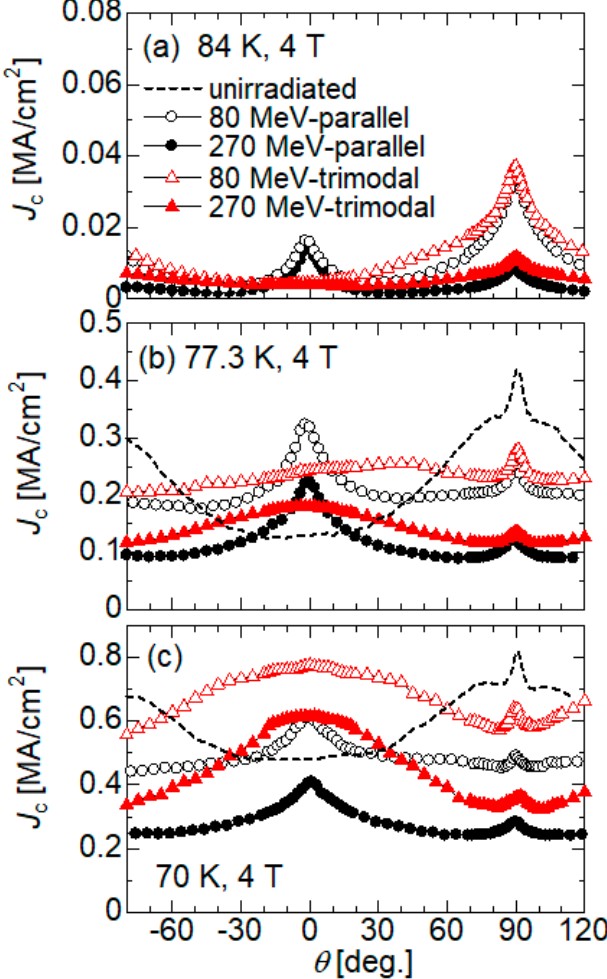

**Figure 21.** Magnetic field angular dependence of $J_c$ at magnetic field of 4 T and temperatures of (**a**) 84 K, (**b**) 77.3 K, and (**c**) 70 K for GdBCO coated conductors irradiated with 80 MeV and 270 MeV Xe ions, where the irradiation angles are $\theta_i = 0°$ for parallel CD configurations and $\theta_i = 0°$, $\pm 45°$ for trimodal angular configurations, respectively. The broken lines for (**b**) 77.3 K and (**c**) 70 K show the $J_c$ properties of the unirradiated sample as reference data. Reprinted with permission from [50], copyright 2020 by the Japan Society of Applied Physics.

At the intermediate angles between $B \| c$ and $B \| ab$, on the other hand, the continuous CDs crossing at $\theta_i = \pm 45°$ predominantly trap flux lines. In addition, the flux pinning of CDs crossing at $\theta_i = \pm 45°$ is further enhanced through the suppression of the motion of kinks of flux lines by the gaps in discontinuous CDs, as shown in Figure 22b. Thus, the hybrid flux pinning by the two different kinds of PCs causes the large enhancement of $J_c$ even at the intermediate magnetic field angles.

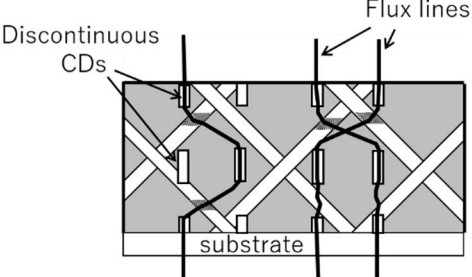
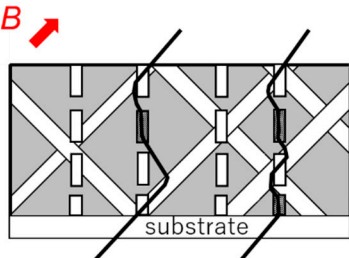

(a) Magnetic field parallel to the *c*-axis    (b) Magnetic field tilted off the *c*-axis

**Figure 22.** Schematic images of flux pinning for the 80 MeV trimodal angular configuration in a magnetic field *B* (**a**) along the *c*-axis and (**b**) in the intermediate angular region between $B \parallel c$ and $B \parallel ab$. The hatched regions in the inclined CDs and the discontinuous ones represent the interaction areas with kinks of flux lines. Reprinted with permission from [50], copyright 2020 by the Japan Society of Applied Physics.

There is a possibility that direction-dispersed CDs with "complete discontinuity" further provide a high and isotropic $J_c$ in high-$T_c$ superconductors. In fact, BaHfO$_3$ nanorods tend to grow discontinuously and to be widely dispersed in the directions, causing a significant improvement of $J_c$ in a wide magnetic field angular range for REBCO thin films [55,56]. The irradiation using lighter ions with lower energy, which provides lower $S_e$ for high-$T_c$ superconductors (e.g., Kr-ion irradiation with 80 MeV, where $S_e$ = 16.0 keV/nm), may produce discontinuous CDs even in directions tilted from the *c*-axis. However, there is a trade-off between the discontinuity of CDs and the thickness of CDs for the formation of CDs by heavy-ion irradiations: discontinuous CDs tend to be thin diameter [18,25], where the elementary pinning force of one segmented column with thin diameter becomes weak. Thus, the discontinuity of CDs does not always provide the strong pinning landscape for the heavy-ion irradiation process. The introduction of direction-dispersed, discontinuous, and thick CDs by the ion irradiation process can be the key to further making high $J_c$ fairly isotropic in high-$T_c$ superconductors.

## 4. Conclusions

We have systematically examined the modification of the anisotropy of $J_c$ in REBCO thin films by using heavy-ion irradiations: the morphology and the configuration of the irradiation defects were controlled by the irradiation conditions such as the irradiation energy and the incident direction. The direction-dispersed CDs were designed in REBCO thin films to push up the $J_c$ in the magnetic field angular region from $B \parallel c$ to $B \parallel ab$, by controlling the irradiation directions. When the directions of CDs were extensively dispersed around the *c*-axis, the $J_c$ was enhanced over a wider magnetic field angular region centered at $B \parallel c$. The $J_c$ at $B \parallel ab$, on the other hand, was hardly affected even by CDs tilted toward the *ab*-plane, which is attributed to the strong line tension energies of flux lines around $B \parallel ab$ in the anisotropic superconductors. We demonstrated the improvement of $J_c$ at $B \parallel ab$ by the introduction of CDs, where the angle of CDs relative to the *ab*-plane were controlled down to $\Theta_i = 5°$. These results suggest that direction-dispersed CDs can provide the isotropic enhancement of $J_c$ over all magnetic field angular region when the angles of CDs are matched with the anisotropic line tension energy of flux lines.

Another promising morphology of CDs, i.e., discontinuous CDs, which can be introduced by heavy-ion irradiation with relatively low energy, showed large potential for the enhancement of $J_c$ over a wide magnetic field angular region. In particular, the combination of the discontinuity and the direction-dispersion lead to further enhancement of $J_c$: the gaps in discontinuous CDs provide the suppression of the motion of flux lines, while the direction-dispersion of CDs produces the strong flux pinning over a wide magnetic field angular region.

The heavy-ion irradiation to high-$T_c$ superconductors can provide the flux pinning structure with higher and more isotropic $J_c$, by further tuning the irradiation process: the systematic studies using the ion irradiation process may lead to the approach to the theoretical limit of $J_c$, i.e., pair-breaking critical current density.

**Author Contributions:** Conceptualization, T.S. All authors have read and agreed to the published version of the manuscript.

**Funding:** This work was supported by KAKENHI (25420292, 16K06269 and 19K04474) from the Japan Society for the Promotion of Society.

**Data Availability Statement:** The data presented in this study are available on request from the author.

**Acknowledgments:** The author thanks N. Ishikawa, M. Mukaida, A. Ichinose, K. Yasuda, T. Fujiyoshi, S. Semboshi, T. Ozaki, and H. Sakane for their constant support during the researches. Experimental works were partly performed under the Common-Use Facility Program of JAEA.

**Conflicts of Interest:** The authors declare no conflict of interest.

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
