# Peer review of "Modification of Critical Current Density Anisotropy in High-Tc Superconductors by Using Heavy-Ion Irradiations"

_qubs, doi:10.3390/qubs5020016_

Round 1
Reviewer 1 Report
The author summarises his previously published work about the effect of heavy ion irradiation at different incidence angle combinations on the anisotropy of the critical current density in YBCO and GdBCO.
The summary is well written except for a few minor shortcomings.
However, the manuscript falls short as a review article as it does not include the work of other authors (e.g. on other high-Tc materials and/or other ion species), lacks aspects such as the dependence on fluence or the effects on the critical temperature, etc.
Reviewer 2 Report
The author reviews the recent results about the modification of the Jc properties in
REBCO thin films and coated conductors were obtained by using heavy-ion irradiations under various irradiation conditions. The topic is very interesting and the manuscript discusses deeply the science behind it. I have small questions regarding their assumptions. However, overall, I believe that this manuscript is a valuable contribution to the literature and, once my comments below are adequately addressed, should be published.

Round 2
Reviewer 1 Report
The manuscript is a well written summary of the authors previous work and all questions have been addressed sufficiently.
However, I would expect a more broad overview in a review paper.
